# Assessing Biocompatibility of Face Mask Materials during COVID-19 Pandemic by a Rapid Multi-Assays Strategy

**DOI:** 10.3390/ijerph18105387

**Published:** 2021-05-18

**Authors:** Tiziana Petrachi, Francesco Ganzerli, Aurora Cuoghi, Alberto Ferrari, Elisa Resca, Valentina Bergamini, Luca Accorsi, Francesco Burini, Davide Pasini, Gaelle Françoise Arnaud, Mattia Piccini, Laura Aldrovandi, Giorgio Mari, Aldo Tomasi, Luigi Rovati, Massimo Dominici, Elena Veronesi

**Affiliations:** 1Technopole “Mario Veronesi”, via 29 Maggio, 6, 41037 Mirandola, Italy; tiziana.petrachi@tpm.bio (T.P.); francesco.ganzerli@tpm.bio (F.G.); aurora.cuoghi@tpm.bio (A.C.); alberto.ferrari@tpm.bio (A.F.); elisa.resca@tpm.bio (E.R.); valentina.bergamini@tpm.bio (V.B.); luca.accorsi@tpm.bio (L.A.); francesco.burini@tpm.bio (F.B.); Davide.pasini@tpm.bio (D.P.); gaelle.arnaud@tpm.bio (G.F.A.); mattia.piccini@tpm.bio (M.P.); laura.aldrovandi@tpm.bio (L.A.); giorgio.mari@tpm.bio (G.M.); aldo.tomasi@unimore.it (A.T.); luigi.rovati@unimore.it (L.R.); massimo.dominici@unimore.it (M.D.); 2Department of Engineering “Enzo Ferrari”, University of Modena and Reggio Emilia, via Vivarelli, 10, Building 26, 41124 Modena, Italy; 3Department of Diagnostic and Clinical Medicine and Public Health, University of Modena and Reggio Emilia, via del Pozzo, 71, 41124 Modena, Italy; 4Department of Medical and Surgical Sciences for Children & Adults, University of Modena and Reggio Emilia, via del Pozzo, 71, 41124 Modena, Italy

**Keywords:** biocompatibility, cytotoxicity, cytokines, inflammation, materials

## Abstract

During the coronavirus disease 2019 (COVID-19) pandemic, scientific authorities strongly suggested the use of face masks (FMs). FM materials (FMMs) have to satisfy the medical device biocompatibility requirements as indicated in the technical standard EN ISO 10993-1:2018. The biologic evaluation must be confirmed by in vivo tests to verify cytotoxicity, sensitisation, and skin irritation. Some of these tests require an extensive period of time for their execution, which is incompatible with an emergency situation. In this study, we propose to verify the safety of FMMs combining the assessment of 3-[4,5-dimethylthiazolyl-2]-2,5-diphenyltetrazolium bromide (MTT) with quantification of nitric oxide (NO) and interleukin-6 (IL-6), as predictive markers of skin sensitisation or irritation based on human primary fibroblasts. Two hundred and forty-two FMMs were collected and classified according to spectrometer IR in polypropylene, paper, cotton, polyester, polyethylene terephthalate, 3-dimensional printing, and viscose. Of all FMMs tested, 50.8% passed all the assays, 48% failed at least one, and only 1.2% failed all. By a low cost, rapid and highly sensitive multi assays strategy tested on human skin fibroblasts against a large variety of FMMs, we propose a strategy to promptly evaluate biocompatibility in wearable materials.

## 1. Introduction

In April 2020, the World Health Organisation declared the coronavirus disease 2019 (COVID-19) pandemic, caused by the SARS-CoV-2 virus. Social contact increases infection rates due to the spread of saliva droplets in the air and on surfaces through coughing and sneezing. Various measures, such as better sanitation, social distancing, and the use of face masks (FMs), were recommended by the health authorities to reduce the spread of the infection [1].

The massive general use of FMs resulted in a global supply shortage of these devices, in particular in health care settings. The difficulties of FM supply and distribution prompted several companies to convert their manufacturing to FM production. Those companies in Italy were certified by the Italian health authority according to 17 March 2020 n.18 (art. 15) law decree. FMs had to satisfy the medical device performance requirements (according to EN 14683:2019+AC, [2]) and biocompatibility requirements as indicated in the technical standard EN ISO 10993-1:2018 [3]. In this context, the law recommends that biologic evaluation had to be confirmed by means of laboratory tests to verify cytotoxicity, sensitisation, and skin irritation.

In particular, irritation and sensitisation are routinely studied through in vivo assays defined as primary skin irritation and guinea pig maximisation tests. Regrettably, these tests are based on living beings with a response time of up to 6 weeks. In compliance with the Italian law decree of 17 March 2020 n. 18 (art. 15), alternative in vitro tests for irritation and sensitisation effects were proposed in an effort to rapidly produce biocompatible FMs. On this regard, ISO TR 15499:2016 (§6.3 Device Testing Consideration, [4]) recommends the adoption of a phased approach with the execution of an in vitro test first to reduce the use of in vivo assays, hence to limit the use of resources and animals. ISO TR 15499:2016 [4] has been completely incorporated into ISO 10993-1:2018 [3] both of which strongly encourage the use of scientific advances in understanding basic mechanisms in order to minimise the number and the exposition of test animals by giving preference to in vitro models and to chemical, morphological, and topographical characterisation testing, in situations where these methods yield equally relevant information to that obtained from in vivo models. Furthermore, as indicated in ISO 14971:2020 [5], if health risks for the people in contact with the medical device are low by scientific evidence, then the devices do not require further mitigation measures. 

Previous investigations assessed biocompatibility through in vitro tests on eight commercially available surgical masks combining cytotoxicity by MTT (3-[4,5-dimethylthiazolyl-2]-2,5-diphenyltetrazolium bromide) assay and nitrite measurement [6].

Several studies have additionally evaluated the capacity of human primary skin cells (i.e., keratinocytes, fibroblasts, melanocytes, and endothelial cells) to express nitric oxide synthase (NOS) and release nitric oxide (NO) [7]. Notably, fibroblasts are capable of spontaneous production of NOS and NO, especially in the presence of an appropriate stimulus, such as lipopolysaccharides (LPSs) or Triton X, a skin irritant. Owing to its short half-life, NO is instantaneously converted and stored in the form of nitrite [8]. Therefore, the nitrite measurement by a Griess assay appears to be a valuable method to investigate irritation and sensitisation in vitro [9]. 

Interleukin 6 (IL-6) is a glycoprotein produced and secreted by a broad range of cell populations and is considered to be a product of inflammatory response [10].In light of these considerations, to better identify potentially sensitising and irritating compounds, the quantification of IL-6 could be an alternative method to the in vivo tests required by ISO 10993-10 [11]. 

The combination of multiple tests to evaluate the safety of medical devices is supported by a large body of scientific literature [12,13,14] aimed at reducing the need for animal experimentation. Considering these data, a phased strategy was therefore further investigated to test new FMs. On those materials presented for FM manufacturing, we here propose a combinatory multi-test approach able to evaluate the cytotoxicity effect by means of murine L929 cells (as recommended by guidelines and by the literature [6,15,16]) and the irritating/sensitising effect by analysing IL-6 and nitrites expression on human primary fibroblasts. 

## 2. Materials and Methods

### 2.1. Sample Collection

From the 23 of March to the 30 of June 2020, our laboratory received 242 different FM materials (FMMs) from different manufacturers. Using a spectrometer IR (FTIR Spectrometer—Waltham (MA) PerkinElmer, USA), the masks were numbered and classified based on the primary material layer in contact with the skin. Only the FMM in contact with the intact skin was processed for the assays while tying strips and ears loops were omitted. According to the ISO10993-12:2012 [17], materials were incubated in Dulbecco’s modified Eagle’s medium (DMEM, Gibco, Paisley, UK), supplemented with 10% of fetal bovine serum (FBS, Corning, Manassas, VA, USA), and 1% of penicillin/streptomycin (P/S, Gibco, Grand Island, NE, USA) at the ratio of 6 cm^2^/mL for 24 h at 37 °C using gentle agitation. The effective area of the fibre surface per gram can be calculated as (Equation (1)): (1)$$F_{a}=\pi \cdot d\cdot L$$
where *L* is the total length of fibres per gram and *d* the average fibre diameter. 

The fibre density is (Equation (2))
(2)$$\rho_{f}=\frac{4}{\pi \cdot d^{2}\cdot L}$$

Thus, the total length of fibres per gram is (Equation (3))
(3)$$L=\frac{4}{\pi \cdot d^{2}\cdot \rho_{f}}$$

Therefore, substituting this expression of *L* in (1), we can obtain the value of the total area of the sample fibre surface as (Equation (4))
(4)$$TF_{a}=\frac{4}{\rho_{f}\cdot d}\cdot W$$
where *W* is the weight of the sample, and the average diameter *d* was estimated using the AxioZoom V16 microscope (Zeiss, Germany). This approach provided the effective area of the sample fibre surface excluding the empty voxels present inside the material. 

### 2.2. Cytotoxicity Assay

L929 murine fibroblasts (Sigma-Aldrich, St Louis, MO, USA) were seeded according to manufacturer instructions at a density of 15,000/cm^2^ in T75 flasks and cultured with DMEM + 10% FBS + 1% P/S, 2% Glutamine (Gibco, Paisley, UK) at 37 °C/5% CO_2_. Five days after seeding, cells reaching 90% confluence were detached with 0.05% trypsin-EDTA 0.02% (Gibco, Paisley, UK), counted with trypan blue (Sigma-Aldrich, Gillingham, UK) 0.4% for viability exclusion test, seeded into 96-well plates (10,000/100 µL/well) and maintained in culture. After 24 h, cells were stimulated with the extract according to ISO10993-5:2009 [18].

Extracts were filtered to 0.22 µm and diluted at the following concentrations: 100% (Extract100), 46.41% (Extract46.41), 21.54% (Extract21.54), and 10% (Extract10).

Latex (Adventa Health, Kota Bharu, Malaysia) and high-density polyethylene (HDPE, Sigma-Aldrich, St Louis, MO, USA) were used, respectively, as a positive and negative control according to ISO10993-12:2012 [17] and ISO10993-5:2009 [18]. After 24 h of exposure, cells were incubated with 50 µL of MTT solution (Sigma-Aldrich, St Louis, MO, USA) for 2 h at 37 °C. The detection of cell viability was performed by quantifying the optical density at 570 nm using a multi-plate reader spectrophotometer (Enspire, PerkinElmer, Hopkinton MA, USA) after the removal of MTT solution and the subsequent suspension of cells in 100 µL of isopropanol (Sigma-Aldrich, St Louis, MO, USA). The reduction in cell viability compared with the negative control was determined using the following formula (Equation (5)):(5)Cell viability(%)=100×OD570e/OD570b
where *OD570e* is the mean value of the measured optical density of the 100% extract, and *OD570b* is the mean value of the measured optical density of the blanks. The viability of Extract46.41 of the test sample has to be at least the same or higher than that of Extact100. A viability greater than 70% of the blank was evaluated as not cytotoxic. Each experiment was performed six times (technical replicates), as requested by the regulation.

### 2.3. Colorimetric Griess Assay

Primary human foreskin fibroblasts (HFFs; American Type Culture Collection (ATCC), Manassas, VE, USA) were seeded according to manufacturer instructions at a density of 8000 cells/cm^2^ in DMEM containing 10% FBS + 1% P/S + 1% Glutamine. Three days after seeding, the cells were detached with Trypsin/EDTA (0.05%/0.02%), at 37 °C/5% CO_2_, quantified by trypan blue 0.4%, and seeded at a density of 5000 cells with 100 µL per well into 96-well plates maintained in culture for 24 h before beginning the experiment.

HFFs were incubated with Extract100 or culture medium only (negative control) or 1.5% Triton X-100 (positive controls, Sigma-Aldrich, St Louis, MO, USA) for 4 h at 37 °C/5% CO_2_. After incubation, supernatants were collected and mixed with Griess reagent (1% sulphanilamide and 0.1% N-(1-Naphthyl) ethylenediamine dihydrochloride) (Sigma-Aldrich, St Louis, MO, USA) for 10 min at room temperature, according to [6]. The amount of nitrite in the test samples was calculated using a sodium nitrite (Sigma-Aldrich, St Louis, MO, USA) standard curve (range, 100–1.25 µM) and a 570 nm absorbance evaluation using EnSpire (PerkinElmer, Hopkinton, MA, USA). Triton X-100 1.5% was used as the positive control as reported elsewhere [19], while supernatants of untreated cells were considered as the negative control. Experiments were conducted in technical triplicates. 

### 2.4. AlphaLISA

Regarding IL-6 assays, HFFs were incubated with Extract100 or culture medium only (negative control) or 8 µg/mL LPS (positive controls, Sigma-Aldrich, St Louis, MO, USA) for 24 h at 37 °C/5% CO_2_. Supernatants were then collected, and IL-6 quantification was performed by AlphaLISA (PerkinElmer, Hopkinton, MA, USA). A total of 5 µL of the culture medium of each sample was distributed in the wells of the half-area alpha plate (96-multiwell plates) and incubated for 1 h at room temperature with 20 µL of a solution consisting of “acceptor beads” and the anti-analyte antibody. Subsequently, 25 µL of a solution consisting of “donor beads” in AlphaLISA buffer was added to each well and incubated for 30 min in the dark. After this period, the plate was analysed by the EnSpire Plate Reader (PerkinElmer, Hopkinton, MA, USA) instrument with a protocol dedicated to AlphaLISA assays and setting a λ value equal to 615 nm. Experiments were conducted in triplicate. Values were reported as average of expression levels of IL-6 (pg/mL) ± relative standard deviation (RSD%).

### 2.5. Statistical Analysis 

The outcome of the three assays was reported as either “passed” or “failed.” As previously reported for cytotoxicity, the samples were considered to have passed the test if the viability of Extract100 was superior to the viability of 70% of the blanks, as recommended by the ISO10993-5:2009 [18]. For nitrite and IL-6, the averages were compared with respect to the negative controls using a Student *t*-test. Making the null statistical hypothesis (H0) of having no difference between the results obtained from the assays, in triplicate, of the sample and of the negative control, samples were considered to have passed when the assay results were inferior to the negative control or superior but with a *p*-value > 0.05 (H0 true). On the contrary, samples did not pass when their average was superior to the negative control and with a *p*-value < 0.05 (H0 false).

## 3. Results

### 3.1. Face Mask Material Identification and Testing

A total of 242 FMMs were collected and classified according to the type of material used for the internal layer in contact with the skin. When FMs were composed of more than one layer, the internal portion in contact with the skin of the face was used for grouping purposes. Hence, eight main groups were identified, namely: polypropylene (PP), paper, cotton, polyester (PE), polyethylene terephthalate (PET), 3-dimensional (3D) printing, and viscose. Figure 1 depicts a word cloud of the materials composing the FM samples. As shown by font colour and size, the most frequently used materials were PP 52.5% (127 of 242), cotton 19.4% (47 of 242), and polyester 13.2% (32 of 242). The least-numerous groups were a combo of cotton, viscose and elastane 3.7% (9 of 242), PET 3.3% (8 of 242), viscose 3.3% (8 of 242), paper 2.5% (6 of 242), and 3D printing 2.0% (5 of 242).

A total of 726 different tests were performed: 242 MTT assays, 242 tests to quantify IL-6 levels in culture medium, and 242 to evaluate the syntheses of NO. The time required for the execution of the whole protocol, including each of the three assays from cell culture to quantitative analysis using the EnSpire Plate Reader, was reduced to 5 days. 

### 3.2. Face Mask Materials Do Not Reveal Cytotoxicity Showing a Variable Inflammatory Potential 

To characterise the effect of FMs on cell viability according to ISO10993-5:2009 [18] MTT, a cytotoxicity test was performed on L929 cells. As shown in Figure 2a, after 24 h of extract administration, most samples passed, having a viability superior to 70% as recommended by the ISO10993-5:2009 [18]. In detail, more than 90% of the tested FMMs, including (PET, cotton and viscose, PE, PP, and cotton groups) were able to pass the test, while a lower number of FMM samples from viscose, paper and 3D printing were passing.

The inflammatory activity of the FMs was then assessed using Griess assay by measuring the nitrite levels of the supernatant of HFF exposed to Extract100 after 4 h (Figure 2b). Most of the tested FMMs passed the assay, such as 81% of the cotton group, 78% of the cotton and viscose group, and 75–76% of the PP-PET group samples. A lower pass level was observed in the paper and 3D printing samples (57% and 60%, respectively). Overall, the distribution of the Student *t*-test *p*-values obtained for Griess assay was: 180 with *p* > 0.05 (passed); 24 with *p* < 0.001, 20 with 0.001 < *p* < 0.01; 18 with 0.01 < *p* < 0.05 (failed).

Since IL-6 expression is considered an inflammatory marker, we additionally measured this cytokine in HFF supernatant after 24 h of incubation with FMM Extract100. PP group passed the test in 63% of samples, while the cotton and PE groups passed 77% and 73%, respectively (Figure 2c). No irritative response (meaning 100% passed the test) was detected in the cotton, viscose and elastane and the viscose alone groups. Overall, the distribution of the Student *t*-test *p*-values for IL-6 expression obtained was: 170 with *p* > 0.05 (passed); 28 with *p* < 0.001, 22 with 0.001 < *p* < 0.01, 22 with 0.01 < *p* < 0.05 (failed).

In summary, among the 242 tested FMMs, only 2 (1.2%) failed (1 PP, 1 cotton) all the 3 tests, 126 FMMs (50.8% of total) passed them all (61 PP; 3 paper; 28 cotton; 7 cotton, viscose, and elastane; 18 PE; 4 PET; 2 3D printed and 5 viscose; Figure 3a) and the other 120 FMMs (48% of total) failed at least one. Given the high number of FMMs that failed the multi-test approach, we summarised the concordance of three tests in Figure 3b. Remarkably, in 8% of cases, MTT was negative while being positive for IL-6 and Griess (violet bar), showing how materials can stimulate an inflammatory activity despite the lack of cytotoxicity. Similarly, MTT was the only positive test in very few cases (2.8%; green bar), with a positive for Griess and IL-6 in 15.7% (blue bar) and 19.4% (red bar), respectively. These data indicate how Griess and IL-6 represent two distinct markers of inflammation, often resulting in potentially irritating FMM response.

### 3.3. Cytotoxicity, Nitrite, and IL-6 Assays Have Reproducible Results in Three CE-Marked Masks

In order to confirm the reproducibility and predictivity of the described tests, three commercially available and previously (before 17 March 2020 n.18 law decree) CE-marked FMs (named CFM1, CFM2 and CFM3) were taken into account.

Cytotoxicity has been performed according to ISO10993-5:2009 [18], and results are provided in Table 1 for CFM1, CFM2 and CFM3. As recommended, latex, as a highly cytotoxic compound, has been introduced as a positive control. When latex was added to the cultures, the cell viability resulted in <10% (Table 1). On the contrary, HDPE and Extract100 of tested CFM showed a viability of >70%, thus excluding their cytotoxic potential. As expected, the viability of Extract46.41 was found to be similar to or higher than Extract100, satisfying the acceptable criteria recommended by ISO10993-5:2009 [18]. Tested CFM1, CFM2, CFM3 revealed a similar trend in viability, confirming the reproducibility of the approach (Table 1).

Four hours of HFF stimulation using Extract100 on CFM1, CFM2 and CFM3 did not induce a nitrite secretion (value close to 0), and this was also observed in the negative control (*p* > 0.05). On the contrary, Triton X (as positive control) administration increased nitrite levels, confirming the irritant effect of the compound (Table 2).

Concomitantly, the product’s inflammatory potential was assessed by the quantification of IL-6 secretion. As shown in Table 3, we could not detect significant levels of IL-6 in CFM1, CFM2, or CFM3 compared with the negative control (*p* > 0.05). Instead, LPS drastically increased IL-6 secretion, as also reported in [20]. The RSD% of each value was, on average, less than 20% of the expected for biologic samples [21].

These data obtained on three distinct pre-existing CE marked (in vitro and in vivo tested) FMs validate the combinatory MTT test with nitrite and IL-6 quantification, confirming the reproducibility of our assays and their value in rapid and consistent safety screening.

## 4. Discussion

To face the COVID-19 pandemic, scientific evidence and governmental bodies strongly suggested or imposed the use of FMs. Given the massive need for FMs and their global supply shortage, several companies converted their manufacturing operation to produce them. According to international guidelines and based on in vitro and in vivo testing, FMMs had to satisfy the medical devices biocompatibility requirements. In detail they must be neither cytotoxic nor induce skin sensitisation without carrying irritation (ISO10993-1:2018 [3]). However, in vivo tests can take up to 6 weeks, a time that has been considered as significantly long during the COVID-19 emergency. A national law decree of 17 March 2020 n. 18 (art. 15) suggested in vitro tests, only for irritation and sensitisation, in an effort to rapidly test FM biocompatibility.

The aim of the present work was then to identify and share an accurate and predictable in vitro approach able to investigate the safety of FMMs in rapid and easy-to-perform tests.

The here proposed multi-test approach combines cytotoxicity assessment (performed according to ISO10993-5:2009 [18]) with IL-6 and NO quantification as predictive markers of skin sensitisation and irritation on HFFs. This approach reduced the turnaround time to 5 days.

Although in vivo assays are essential for observing the overall effects and for resembling the human microenvironment, ethical issues must be considered. The novel in vitro approach here proposed could contribute to a reduction in animal experimentation in compliance with the principles of the 3Rs (Replacement, Reduction and Refinement) and further emphasised by ISO 10993-1:2018 [3]. The aim is to give preference to in vitro models and to chemical, morphological, and topographical characterisation testing in situations where these methods yield equally relevant information to that obtained from in vivo models [3]. The comparisons of the multi-test approach with in vivo skin irritation and sensitisation assays could be useful to better predict the safety of tested materials. However, international guidelines and the principles of the 3Rs deter the in vivo approach given the high number of animals and the elevated number of evidence in the literature.

Earl, Weil, and Scala suggested that in vivo study should not be the gold standard to validate in vitro test. They reported an extensive inter-laboratories study describing nine different chemicals tested for both eye and skin irritation potential. Experiments were performed in 24 different laboratories. The study reported a considerable bias in the results obtained from tests operated with a common protocol among and within laboratories [22,23]. Therefore, the multi-test approach was validated using three different commercially available CE-marked FMs. As reported, each sample was tested in triplicate and in three different experiments. MTT, nitrite, and IL-6 showed the repeatability and reproducibility of the proposed multi-test approach for all the CE marked FMMs commercially available that have been previously in vivo and in vitro assayed.

In our multi-assay strategy, 48% of the FMs failed the test suggesting the high sensitivity of the approach. These results guarantee the assessment of the safety for FM end-users, predicting the most common skin adverse events such as skin oedema and irritation. Most of the samples that failed the tests reported a high level of inflammation markers rather than cytotoxicity. In detail, the IL-6 test seems to be more sensitive than nitrite evaluation. The high number of analysed FMs allowed us to identify possible responsible factors of the increased concentrations in nitrite and IL-6. These included cleaning approaches applied during or after the manufacturing process to reduce bacterial burden, as recommended by EN14683:2019+AC:2019 [24]. In general, the addition of antibacterial or silver compound in the outside layer of the mask could partially contaminate the inner layer, which can potentially cause an irritation or sensitisation effect. Among others, several antibacterial chemicals, such as hydrogen peroxide, chlorine dioxide, bleach, alcohol, soap solutions, ethylene oxide, ozone decontamination, and physical approaches, such as the use of heat with steam or with dry air, ultraviolet rays, gamma irradiation, or microwaves should be further investigated [25]. For example, although PP was extensively used as medical device components, it often showed irritation and sensitisation potential. We hypothesised that an altered extrusion process could induce many different chemical and physical surface properties, potentially causing the failure of the tests.

Limitations of the study are a lack of clear correlation between nitrite/IL-6 and sensitisation or irritation potential. In fact, these markers indicate an inflammation response of the biological system versus the materials tested. Our approach could be improved with the introduction of further markers or the performance of other in vitro assays such as skin equivalent, as recently introduced in ISO10993-23:2021 [16].

The novelty of the study is the combination of three different tests in order to evaluate more aspects of the biological effect of the material involving cytotoxicity and inflammation. Previously, others suggested the use of in vitro combined tests to predict possible adverse effects of biomaterials [12,13,14]. In particular, most of these inflammatory assays are based on the use of murine macrophages and monocytes [12,13]. The novelty of our approach, which significantly differs from those, is the introduction of human primary fibroblasts with a better predictivity of the assay [26]. Human sources allow the reduction in inter-species variability. In addition, skin fibroblasts were selected, on one side, for their critical and underappreciated role in the switch from acute inflammation to adaptive immunity [27] and, on the other, considering this cell population representative of the skin and their fibroblasts, as the first tissue on which FMMs come into contact [28].

The use of skin specific cells within in vitro multi-test combination represents an alternative, rapid and more representative strategy to quickly predict the possible toxic potential of FMMs.

## 5. Conclusions

The use of skin specific cells within in vitro multi-test combinations represents an alternative, rapid, and more representative strategy to quickly predict the possible toxic potential of FMMs. In the fight against any pandemic, including COVID-19, these features are essential in order to effectively test were able and disposable devices to prevent or limit infectious spread. Moreover, our study could be useful not also for face mask characterisation, but it could be successfully extended to evaluate the safety of other materials commonly involved in medical device manufacturing.

## Figures and Tables

**Figure 1 ijerph-18-05387-f001:**
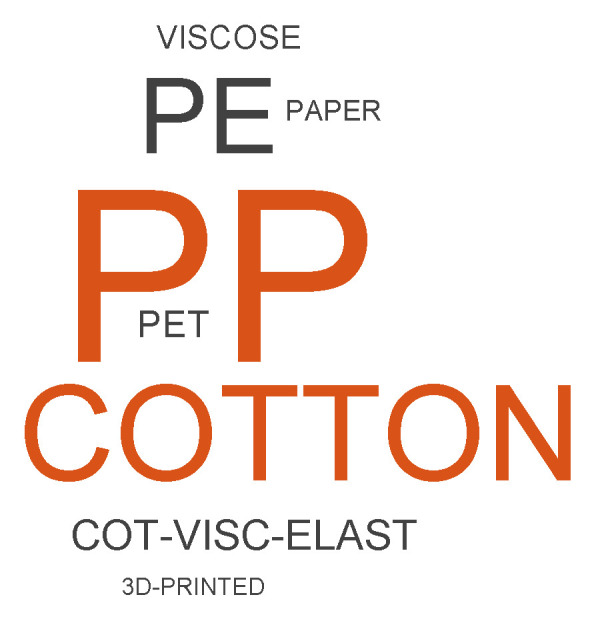
Face Mask Materials. Word cloud of the main material composing the samples: 127 of 242 were polypropylene (PP); 47 of 242 cotton; 32 of 242 polyester (PE); 9 of 242 cotton/viscose/elastane; 8 of 242 PET; 8 of 242 viscose; 6 of 242 paper; and 5 of 242 3D printing.

**Figure 2 ijerph-18-05387-f002:**
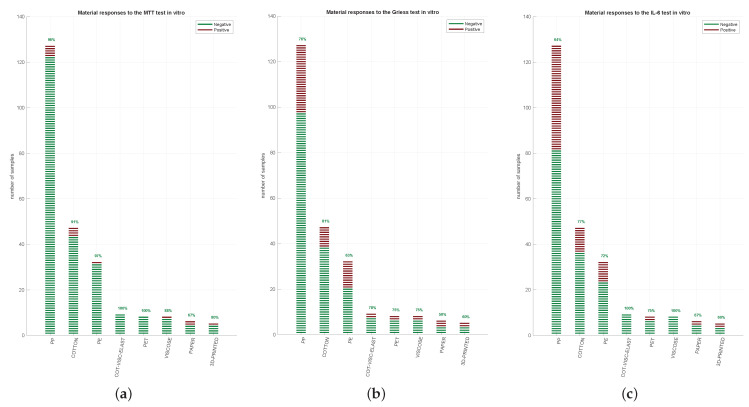
Single assay results. 242 FMMs were divided into 8 groups (PP, cotton, PE, a combo of cotton, viscose and elastane, PET, viscose, paper, and 3D-printed). Each material was analysed by MTT (**a**), Griess’ test (**b**) and IL-6 quantification (**c**). The green lines represent the samples that “passed” the test, red lines indicate the samples that “failed” the test. In regard to cytotoxicity, the samples were considered to have passed if the viability of Extract100 was superior to 70% as recommended by the ISO10993-5:2009, whereas for nitrite and IL-6, the averages were compared with negative controls using a Student *t*-test (Section 2.5 Statistical Analysis).

**Figure 3 ijerph-18-05387-f003:**
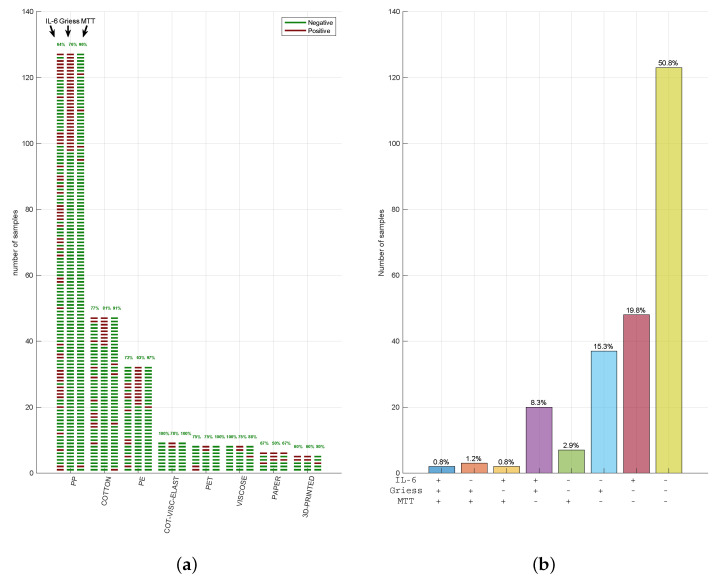
Face mask material response in our multi-assays strategy. (**a**) Out of the 726 different assays performed (242 MTT, 242 IL-6 levels, and 242 syntheses of NO), raw representation of the responses matching the cases in which the 242 samples evaluated were simultaneous negative or positive in all 3 tests. The green lines represent the samples that “passed” the test, while red lines indicate the samples that “failed” the test. (**b**) Summary of the concordance of results. The percentage resulting from the combination of the outcome, “passed” (−) or “failed” (+), of the 3 different assays performed is represented.

**Table 1 ijerph-18-05387-t001:** Relative cell viability (% of blank) of three single internal parts of certified face masks (CFM1,2,3) measured by MTT assay.

CFM1							
	Blank	HDPE	Latex	Extract100	Extract46.41	Extract21.54	Extract10
PART#1	100	90.86 ± 14.04	6.54 ± 0.39	95.28 ± 14.22	97.05 ± 12.68	98.39 ± 16.92	98.25 ± 16.57
PART#2	100	93.26 ± 11.15	5.66 ± 0.38	90.58 ± 9.93	93.23 ± 15.52	93.32 ± 9.70	94.59 ± 7.10
PART#3	100	92.14 ± 6.58	8.45 ± 0.34	94.68 ± 10.33	98.08 ± 9.03	101.39 ±10.13	101.46 ± 7.87
**CFM2**							
	Blank	HDPE	Latex	Extract100	Extract46.41	Extract21.54	Extract10
PART#1	100	90.79 ± 3.64	6.23 ± 0.49	90.68 ± 5.52	91.57 ± 9.37	92.19 ± 9.03	92.56 ± 6.87
PART#2	100	95.44 ± 6.65	6.04 ± 0.43	97.22 ± 9.97	98.43 ± 11.78	100.59 ± 6.51	100.64 ± 4.24
PART#3	100	87.39 ± 6.65	5.54 ± 0.37	9.87 ± 6.66	97.04 ± 3.76	97.37 ± 6.17	97.98 ±12.12
**CFM3**							
	Blank	HDPE	Latex	Extract100	Extract46.41	Extract21.54	Extract10
PART#1	100	87.38 ± 5.54	5.53 ± 0.34	84.26 ± 4.93	84.92 ± 2.42	85.87 ± 4.46	86.22 ± 3.50
PART#2	100	90.83 ± 6,12	6.59 ± 0.52	93.20 ± 4.51	95.99 ± 6.47	96.09 ± 7.62	96.26 ± 3.10
PART#3	100	92.11 ± 14.90	6.82 ± 0.47	92.78 ± 12.43	102.52 ± 14.06	110.19 ± 6.76	111.49 ± 12.47

**Table 2 ijerph-18-05387-t002:** Nitrite levels in the supernatant of three single internal parts of CFM1, CFM2 and CFM3 after exposure to Extract100 for 4 h. Results shown are as average values ± SD (%).

	AVERAGE (µg/mL) ± SD%		
Negative Control	−0.98 ± 0		
Triton X-100	115.69 ± 0.09		
	part#1	part#2	part#3
CFM1	−0.98 ± 0	−0.98 ± 0	−0.86 ± 0.09
CFM2	−0.70 ± 0.09	−0.98 ± 0	−0.92 ± 0.09
CFM3	−0.92 ± 0.09	−0.92 ± 0.09	−0.89 ± 0.09

**Table 3 ijerph-18-05387-t003:** IL-6 levels in the supernatant of three single internal parts of CFM1, CFM2, and CFM3 after exposure to Extract100 for 24 h. Results shown are as average values ± RSD (%).

	AVERAGE (pg/mL) ± RSD%		
Negative Control	116.99 ± 10.51		
Lipopolysaccharide	897.63 ± 0.60		
	part#1	part#2	part#3
CFM1	133.06 ± 2.94	124.85 ± 12.58	104.41 ± 15.32
CFM2	97.70 ± 3.00	90.79 ± 12.07	89.38 ± 12.99
CFM3	123.72 ± 12.28	120.03 ± 2.90	104.19 ± 6.58

## Data Availability

The data presented in this study are available in this article.

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
