# Peer review of "Assessing Biocompatibility of Face Mask Materials during COVID-19 Pandemic by a Rapid Multi-Assays Strategy"

_ijerph, 2021, doi:10.3390/ijerph18105387_

Round 1

Reviewer 1 Report

This study is for the biological safety evaluation about the face mask. Therefore, in addition to evaluation through cell culture, additional evaluation is required according to the Iso - 10993 evaluation criteria.

The evaluation shall be performed animal testing sucah as skin irritation or sensitization.

In addition, the virus penetration evaluation is very important for coronavirus mask evaluation.

So, the important assessment is transmittance test shall be carried out for nano or micro particles.

Reviewer 2 Report

This research is under the scope of this journal; the topic is relevant for readers, and this research deals with potentially significant knowledge to the field.

However, there are some concerns about the present manuscript: 

Abstract

• How many samples? Identified in the abstract.

• The authors should describe how the results were expressed and statistical analysis performed. 

• In the results, is important to show more information, add some of the p-values.

Introduction:

This not adequate for the introduction section. Whole of discussion is poorly written and insufficient. What is the importance of this study? 

Please consider this reference about cytotoxicity: doi: 10.1111/jopr.13226. Epub 2020 Jul 30

Please add the null hypothesis in the aim section.

Material and methods: Number of samples. The authors should be clarifying this in each experiment.

Lack of significance (p <???) in the explanations, in the results section

There are many mistakes in the references section and in the text 

The discussion is also misleading. What is the novelty of this paper???

Limitations? Consider the use of human cells: doi: 10.1007/s10856-013-4849-x. E

Conclusions were not totally supported by the data showed.

Figure legends: Bad descriptions

Author Response

How many samples? Identified in the abstract.

We thank the reviewer for pointing this out. We agree on the opportunity to add this information into the abstract. Sample number was added into the abstract line 28.

The authors should describe how the results were expressed and statistical analysis performed. 

We thank the reviewer for this comment. The Materials and Methods section contains the 2.5 paragraph dedicated to data analysis and result reporting.

This paragraph has been revised in order to increase clarity on presentation of the data elaboration performed.

Section 2.5 of Material and Methods has been revised in order to increase the level of detail and information regarding data elaboration and the statistical analysis performed

In the results, is important to show more information, add some of the p-values.

This is a well-taken point and we agree with the reviewer on the opportunity to add p-values where Student t test was performed (nitrite and IL-6).

Result section was updated with the information regarding the statistical distribution of the p-values obtained from the assays evaluations. Please see Lines 240- 242 and lines 247-249.

This is not adequate for the introduction section. Whole of discussion is poorly written and insufficient. What is the importance of this study? 

  • Please consider this reference about cytotoxicity: doi: 10.1111/jopr.13226. Epub 2020 Jul 30
  • Please add the null hypothesis in the aim section. Material and methods: Number of samples. The authors should be clarifying this in each experiment. Lack of significance (p <???) in the explanations, in the results section

We thank the reviewer for the useful and punctual suggestion on improvement of the quality of text.

The suggested references were added.

The null hypothesis was declared in section 2.5 of Materials and Methods.

The number of experiments performed and samples analysed is indicated in the Result section line 213 and 229. To increase the visibility of this information, it has been added also in the caption of figure 2 and 3.

The two references suggested by the reviewer were added in the introduction [15 and 16].

The null hypothesis was defined in section 2.5 of Materials and Methods.

Caption of figures 2 and 3 have been revised to provide more emphasis on samples and assays performed

There are many mistakes in the references section and in the text 

We thank the reviewer for the useful review. References has been accurately checked on their correspondence on text and on their correctness in the Bibliography section.

Bibliography and text has been revised on order to correct mistakes in references.

The discussion is also misleading. What is the novelty of this paper???

Limitations? Consider the use of human cells: doi: 10.1007/s10856-013-4849-x. E

Dear Reviewer, thank for your suggestion, we improved the discussion emphasizing the novelty of the paper and the future use of the multi-test approach.

We also added a sentence about the limitations regard to the quantification of nitrite and IL-6 for irritation/sensitization response in lines 376-380. 

New paragraphs were inserted into the discussion in order to emphasise the novelty (please see lines 284-395) and limitations (please see lines 376-380) of the paper.

References 16 and 26 were added in the references section.

Conclusions were not totally supported by the data showed.

This is a well-taken point that helped us to improve the conclusion section.

Conclusion section was revised and improved (please see lines 406-408).

Figure legends: Bad descriptions

This is a well-taken point that helped us to improve the captions of the figures. Not only, we have also improved the readability of the figures by presenting assays results in a decreasing order of materials group dimension.

Figures and Figures  captions were revised and improved.

Reviewer 3 Report

This is an interesting contribution on the use of in vitro methods to replace in vivo testing. The authors focus on face mask materials, but in my opinion the approach is applicable to other materials as well. This issue should be more highlighted in the body text and should even receive more attention instead of focussing on face masks, the more as the first countries in the world have stopped the obligation of wearing face masks.

A second issue relates to the fact that more attention should be paid to comparing in vitro test results to in vivo data. This aspect is mentioned, but should be further dealt with in the abstract, body text, and conclusions section.

A minor issue: the abbreviation MTT in the abstract should be given in full.

Author Response

This is an interesting contribution on the use of in vitro methods to replace in vivo testing. The authors focus on face mask materials, but in my opinion the approach is applicable to other materials as well. This issue should be more highlighted in the body text and should even receive more attention instead of focussing on face masks, the more as the first countries in the world have stopped the obligation of wearing face masks

This is a well-taken point that helps us to broaden the discussion on the results obtained. The authors also strongly support and foster where possible and appropriate the use of in vitro methods to replace in vivo testing.

We are confident that this issue could greatly enriched the meaning of the manuscript.

A sentence was added in conclusion at line 406-408 promoting the in vitro test here proposed for face masks to be implemented also for other materials and applications.

A second issue relates to the fact that more attention should be paid to comparing in vitro test results to in vivo data. This aspect is mentioned, but should be further dealt with in the abstract, body text, and conclusions section

The comment is indeed very appropriate and we extensively discuss about comparison with in vitro and in vivo test in the manuscript.

paragraphs were added in introduction section (please see lines 58-66) and in discussion section (please see lines 330-351).

A minor issue: the abbreviation MTT in the abstract should be given in full.

We thank the reviewer for pointing it out. The abstract was updated according to the suggestion of the reviewer and adding “3-[4, 5-dimethylthiazolyl-2]-2, 5-diphenyltetrazolium bromide” close to the abbreviation MTT in the abstract section.

The abstract was updated according to the suggestion of the reviewer.

Round 2

Reviewer 1 Report

I saw the author's answer. However, these studys show a great lack of academic and scientific aspects. Therefore, it is judged that it is more of a general verification than an academic paper.
